# High-Frequency Surface Acoustic Wave Resonator with Diamond/AlN/IDT/AlN/Diamond Multilayer Structure

**DOI:** 10.3390/s22176479

**Published:** 2022-08-28

**Authors:** Liang Lei, Bo Dong, Yuxuan Hu, Yisong Lei, Zhizhong Wang, Shuangchen Ruan

**Affiliations:** College of Integrated Circuits and Optoelectronic Chips, Shenzhen Technology University, Shenzhen 518118, China

**Keywords:** SAW, sandwiched IDT, multilayer SAW resonator, high frequency, FEM

## Abstract

A high-frequency surface acoustic wave (SAW) resonator, based on sandwiched interdigital transducer (IDT), is presented. The resonator has the structure of diamond/AlN/IDT/AlN/diamond, with Si as the substrate. The results show that its phase velocity and electromechanical coupling coefficient are both significantly improved, compared with that of the traditional interdigital transduce-free surface structure. The M2 mode of the sandwiched structure can excite an operation frequency up to 6.15 GHz, with an electromechanical coupling coefficient of 5.53%, phase velocity of 12,470 m/s, and temperature coefficient of frequency of −6.3 ppm/°C. This structure provides a new ideal for the design of high-performance and high-frequency SAW devices.

## 1. Introduction

A surface acoustic wave (SAW) is a wave whose energy is only concentrated near the surface of an elastic body [1]. It has many advantages. First, its propagation speed is five orders of magnitude smaller than that of the electromagnetic waves, which can not only reduce the size and weight of electronic devices, but also greatly improve their performance [2]. Second, the signal can be accessed during the propagation of the acoustic surface wave [3]. Third, it is easy to mass-produce SAW devices with reliable performance and good consistency using the semiconductors manufacturer technology. Because of the above reasons, various SAW devices can be quickly developed and put into practical application, Figure 1 shows the basic structure during a common SAW.

With the rapid development of information and communication technology, the frequency of SAW devices is increasing continuously, from a few MHz at the beginning to a few GHz at present. In terms of both the development of SAW technology and the application requirements of SAW devices, the development direction of SAW devices is to develop to high-frequency and high performance [5,6]. Therefore, many researchers are working to develop a series of innovative devices with high frequency and good performance [7,8,9]. For SAW device, the frequency is determined by *f* = $v_{p}$*/λ*, where $v_{p}$ is the SAW phase velocity of the substrate and *λ* is the acoustic wavelength, as determined by the IDT periodicity [10]. Since the SAW speed of traditional piezoelectric single crystal substrate materials, such as LiTaO_3_, is only about 4000 m/s and the limit width of the applicable forkfinger electrode (IDT) is about 0.25 μm, the service life, antistatic ability, and power tolerance of the electrode will be seriously decreased, and its ohmic loss will also be significantly increased. Compared with piezoelectric single-crystal materials, piezoelectric multilayer SAW devices composed of multiple media have attracted more attention in recent years [11]. The SAW propagation characteristics are not only related to the acoustic propagation characteristics of the piezoelectric film and substrate materials themselves, but also depend on the thickness of the piezoelectric film and the excitation mode of SAW [12]. Selecting the appropriate material and film thickness and using the appropriate excitation method can make the acoustic properties of the piezoelectric multilayer film medium better than that of the piezoelectric single crystal material, which is of great significance for improving the performance of the device [13].

Diamond has the highest $v_{p}$ and thermal conductivity, 2.5–3 times that of traditional materials [14], but it has no piezoelectric properties and needs to be deposited on the piezoelectric thin film materials for SAW device application [15,16]. AlN is the material with the highest $v_{p}$ among the known piezoelectric materials [17,18]. In 2005, M. Benetti et al. calculated the SAW characteristics of AlN/diamond under four excitation modes and demonstrated the IDT/AlN/diamond/Si multilayer device, but the loss is relatively large, and the electromechanical coupling coefficient ($K^{2}$) is not more than 1% [19]. In 2017, Lei Wang proposed the AIN/diamond/silicon layered substrate structure with an embedded IDT [5]; the pseudo-surface wave was used to achieve a maximum resonance frequency of 17.7 GHz with a cross-finger width of 0.125 um and $K^{2}$ of 0.92%. It is difficult to apply in actual industrial production. In 2020, Yusuke Kobayashi et al. deposited ScAIN thin films on polycrystalline diamond (PCD) and heteroepitaxial diamond (HED) substrates [20]. IDTs of 0.8 um and 0.5 um were fabricated and achieved a $K^{2}$ of 5.40~5.52% and SAW $v_{p}$ of 7400~7766 m/s. However, the operating frequency is from 2.2 to 3.5 GHz. Therefore, making the IDT finger width in the range of industrial production and achieving a higher operating frequency have become the focus of the research of layered SAW devices.

Here, a diamond/AlN/IDT/AlN/diamond-based SAW resonator with high-frequency resonance up to 6.15 GHz is presented. The resonator adopts Si material as the substrate structure, and two layers of diamond film and two layers of AlN film are attached. IDT is sandwiched in the piezoelectric layer. The diamond/AlN/IDT/AlN/diamond structure is shown in Figure 2. The new design is analyzed by using the finite elements method (FEM). Simulated results show that its $K^{2}$ can reach 5.53%; correspondingly, $v_{p}$ reaches 12,470 m/s, both of which are much higher than that of the traditional design. Moreover, the structure can easily excite high-frequency resonance up to 6.15 GHz. This new design provides us a good solution to achieve high-performance and high-frequency SAW devices.

## 2. Establishment and Simulation of SAW Resonator

The piezoelectric effect is generated by the interaction between Coulomb force and elastic restoring force. The piezoelectric constitutive relation describes the interaction between electrical parameters (electric field strength and electric displacement) and mechanical parameters (stress and strain) in piezoelectric media, which can be expressed as follows [21]:(1)$$T_{ij}=c_{ijkl}^{E}S_{ij}-e_{kij}E_{k},$$
(2)$$D_{i}=e_{ikl}S_{kl}+\varepsilon_{ij}^{S}E_{j},$$
where $T_{ij}$ is stress, $D_{i}$ is electric displacement, *S* is strain, *E* is electric field intensity, $c_{ijkl}^{E}$ is the elastic constant, $e_{kij}$ is the relative dielectric constants, and $\varepsilon_{ij}^{S}$ is the dielectric constant.

The propagation speed of the surface acoustic wave is 4~5 orders of magnitude lower than that of the electromagnetic wave. Therefore, the electromagnetic field, coupled with the surface acoustic wave, can be approximated as the electrostatic field, and the intensity of the electric field can be expressed as the gradient of an electric potential function [22]:(3)$$E_{i}=-\frac{\partial \varphi }{\partial x_{i}}$$

From the Gaussian equation, considering that there is no free charge in the insulating medium, the divergence of the electric displacement vector D is zero [23]:(4)$$\frac{\partial D_{i}}{\partial x_{i}}=0$$

Substituting Equations (3) and (4) into Equation (1) and combining the law of conservation of momentum and Gauss’s theorem, the general partial differential equation of the quasi static state of the piezoelectric device is expressed by [24]:(5)$$c_{ijkl}^{E}\frac{\partial^{2}u_{k}}{\partial x_{i}\partial x_{l}}+e_{kij}\frac{\partial^{2}\varphi }{\partial x_{i}\partial x_{l}}=\rho \frac{\partial^{2}u_{i}}{\partial t^{2}},$$
(6)$$e_{ikl}\frac{\partial^{2}u_{k}}{\partial x_{l}\partial x_{i}}-\varepsilon_{ik}^{S}\frac{\partial^{2}\varphi }{\partial x_{k}\partial x_{i}}=0,$$

In the Equations (3) and (4), $u_{i,k}$ is the displacement of the particle, *t* is time, *ρ* is density, *φ* is potential, and $x_{i,j,k,l}$ is different directions.

In this paper, the structural mechanics module of COMSOL Multiphysics software is used to simulate and solve the sandwich structure SAW resonator. For the periodic interfinger electrode with infinite uniform IDT, its surface electric potential $\varphi_{s}$, surface charge density $\delta_{s}$, and tangential component of surface electric field $E_{1}$ all have periodicity [25]. Therefore, periodic boundary conditions and a pair of electrodes in IDT are adopted to simulate the propagation characteristics of the actual multi-layer SAW device, which can reduce the computation time, while not reducing the simulation accuracy. The unit structure model is shown in Figure 3a.

Figure 4 shows the location of the boundary conditions and geometric parameters of the cell structure model, the wavelength *λ* is defined as 2 µm, setting the cross-finger width as 0.25*λ* and cross-finger gap b as 0.25*λ*. The thickness of 3D model in Z direction was set as 0.5*λ*, and the thickness of Si substrate was set as 3*λ*; a perfect matching layer (PML) with a thickness of 0.5*λ* was added to the bottom to reduce the reflection of waves from the bottom boundary. SAW generally propagates in one wavelength, taking the thickness of AlN as *λ*. According to the theoretical analysis by H. Nakahata et al. [26], as long as *k*$h_{\mathrm{dia}}$ > 4 (*k* = 2π/*λ*, $h_{\mathrm{dia}}$ is the thickness of diamond film), the requirement of high sound speed is met; that is, $h_{\mathrm{dia}}$ > 2*λ*/π. The thickness of the diamond film is defined as 0.5λ. Taking the IDT electrode material as an aluminum electrode, the specific geometric parameters are shown in Table 1.

The AlN film used in the simulation is oriented to (002) direction, and the Euler angles are set to be (0°, 0°, 0°). The lower boundary of the model is fixed boundary condition. The front, back, left, and right boundaries are set periodic boundary condition, while the others are free boundaries. One electrode is connected to the ground, and the other are set to be 1 V. See Table 2 for details.

The propagation of SAW in the substrate materials will cause losses. According to the material parameters provided by the Morgan Electro Ceramics (MEC) company, the mechanical $\eta_{\mathrm{CE}}$ and dielectric $\eta_{\varepsilon S}$ losses of most piezoelectric materials are between 0.001 and 0.100 [27]. In this paper, the added $\eta_{\mathrm{CE}}$ and $\eta_{\varepsilon S}$ are 0.010 for AlN piezoelectric material.

After the establishment of the model and setting of boundary conditions, the model needs to be meshed. In order to ensure the accuracy of the calculation, the model is divided, in detail, and the minimum width of the cells is *λ*/20, as shown in Figure 3. After the above setting, the “characteristic frequency” and “frequency domain” studies are added to solve the problem.

According to the simulation steps given above, the material parameters in Table 3 are adopted. The resonant frequency $f_{r}$ and antiresonant frequency $f_{\mathrm{ar}}$ of the first three modes of the two structures are calculated, and their corresponding stress and deformation diagrams are obtained. The low frequency $f_{r}$, corresponding to the intrinsic frequency of the short-circuit electrode, and high-frequency $f_{\mathrm{ar}}$, corresponding to the intrinsic frequency of the open-circuit electrode [12,28]. The $v_{p}$ of SAW can be calculated by the following relation [29]:(7)$$v_{p}=\lambda \frac{\left(f_{r}+f_{\mathrm{ar}}\right)}{2}\;$$

The electromechanical coupling coefficient $K^{2}$ is an important parameter for evaluating the acoustoelectric conversion efficiency, which can be approximately expressed as [30]:(8)$$K^{2}\;\approx \;2\frac{\left(f_{\mathrm{ar}}-f_{r}\right)}{f_{r}}$$

Temperature coefficient of frequency (TCF), which is a measure of the temperature stability for the SAW devices, is obtained by the following equation [31]:(9)$$\mathrm{TCF}=\frac{1}{f_{r}}\frac{\partial f_{r}}{\partial T}\;$$
where *T* is the temperature. At room temperature, Equation (7) can be written as [24,32]:(10)$$\mathrm{TCF}=\frac{1}{40\;{}^{\circ}C-25\;{}^{\circ}C}\frac{f_{r}\left(40\;{}^{\circ}C\right)-f_{r}\left(25\;{}^{\circ}C\right)}{f_{r}\left(25\;{}^{\circ}C\right)}\;$$
where $f_{r}$ (40 °C) and $f_{r}$ (25 °C) are the resonance frequencies at 40 and 25 °C, respectively.

**Table 3 sensors-22-06479-t003:** Material parameters.

	Symbol	AlN [33]	Diamond [19]	Al [34]	Si [35]
Elastic constants (10^11^ N/m^2^)	$c_{11}$	3.45	11.53	1.11	1.66
	$c_{12}$	1.25	0.86	0.59	0.64
	$c_{13}$	1.20	0.86	0.59	0.64
	$c_{33}$	3.95	11.53	1.11	1.66
	$c_{44}$	1.18	5.33	0.26	0.8
Temperature coefficients of elastic constants (10^−4^/°C)	$Tc_{11}$	−0.37	−0.14	−5.9	−0.53
	$Tc_{13}$	−0.018	−0.57	−0.8	−0.75
	$Tc_{33}$	−0.65	−0.14	−5.9	−0.53
	$Tc_{44}$	−0.5	−0.125	−5.2	−0.42
Piezoelectric constants (C/m^2^)	$e_{15}$	−0.48			
	$e_{31}$	−0.58			
	$e_{33}$	1.55			
relative dielectric constants	$\varepsilon_{11}/\varepsilon_{0}$	9.04	5.67	1	10.62
	$\varepsilon_{33}/\varepsilon_{0}$	10.73	5.67	1	10.62
Density (kg/m^3^)	ρ	3.26	3.51	2.7	2.33
Temperature coefficients of mass density (10^−6^/°C)	Tρ	−14.69	−3.6	−1.65	−2.6

## 3. Results and Discussion

Figure 5 shows the input admittance characteristic curve of the electrode of the component structure. As can be seen from the figure, the curves have three peaks, respectively. The first peak is the fundamental mode, denoted M0, second peak is the first mode, denoted M1, and third peak is the second mode (M2). According to the vibration displacement mode analysis in Figure 6, M0 mode is mainly displaced along the X direction, and the displacement along the Z direction is close to zero, indicating that M0 mode is Rayleigh wave, while M2 and M3 are mainly displaced in the Z direction and have no obvious displacement in the X direction, indicating that both modes M1 and M2 are LOVE waves.

The performance of the SAW devices with multilayer membrane structure is not only related to the material properties of transmission medium and its associated transducer electrode structure, but also the thickness of the piezoelectric thin film. With proper selection of materials and film thickness, the acoustic characteristics of piezoelectric multilayer film media will be better, which is of great significance for improving the performance of devices. Therefore, the influence of piezoelectric film thickness on the SAW propagation characteristics of the sandwiched structure will be analyzed below to optimize the geometric structure parameters with better performance.

### 3.1. The Influence of Piezoelectric Film Thickness

Figure 7 shows the curve of $v_{p}$ of M0, M1, and M2 modes changing with $h_{\mathrm{AlN}}$/*λ*. By observing the change of curve characteristics, it can be seen that the $v_{p}$ of the M0, M1, and M2 modes of the sandwiched structure is much higher than that of the traditional acoustic media, and all decrease with the increase of $h_{\mathrm{AlN}}$/*λ*. For the fundamental mode (M0), when $h_{\mathrm{AlN}}$/*λ*→0, the wave velocity is closer to the Rayleigh wave velocity of diamond itself. When $h_{\mathrm{AlN}}$/*λ*→∞, the wave velocity is closer to the Rayleigh wave velocity of AlN film itself. It is worth noticing that for M2 mode, when $h_{\mathrm{AlN}}$/*λ* < 0.5, the M2 $v_{p}$ exceeds the shear volume wave velocity of the diamond film (12,810 m/s), indicating that LOVE wave of M2 mode begins to leak energy to diamond film at this time, and the LOVE wave becomes a leaky surface wave. When $h_{\mathrm{AlN}}$/*λ*→0, it will become the longitudinal body wave velocity (18,000 m/s) of the diamond film. This leakage surface wave is also called a high-velocity pseudo-surface wave (HVPSAW).

SAW propagating in multilayer thin film structure has dispersion; that is, the propagation characteristics of wave are related to $h_{\mathrm{AlN}}$/*λ*. On the premise that $K^{2}$ and $v_{p}$ are large, the dispersion should be reduced as much as possible. Figure 8 shows K^2^ for the first three modes. In general, the overall trend increases first and then decreases with the increase of $h_{\mathrm{AlN}}$/*λ*. For the M0 mode, the overall performance of $K^{2}$ is small and reaches the maximum value around $h_{\mathrm{AlN}}$/*λ* = 1.3, which is 1.48%. However, at this time the $v_{p}$ in Figure 7 is smaller, and the operating frequency is lower. For M1 mode, when $h_{\mathrm{AlN}}$/*λ* = 0.25, its $K^{2}$ reaches the maximum value of 3.4%. However, at this time, the phase velocity in Figure 3 has a large dispersion, which makes the device difficult to manufacture. What is most noteworthy is that the overall performance of M2 mode is relatively high, especially when $h_{\mathrm{AlN}}$/*λ* = 0.9, the maximum value $K^{2}$ is 5.46%. At this time, the $v_{p}$ corresponding to Figure 3 is large, and the dispersion is small, which is suitable for making UHF devices.

Figure 9 shows the curve characteristics of TCF changing with $h_{\mathrm{AlN}}$/*λ*. TCF value is determined by its absolute value and shows stability at different temperatures. Therefore, the overall analysis shows that the TCF of both the M0 and M2 modes increases with the increase of AlN film thickness, while that of M1 mode decreases first and then increases. When $h_{\mathrm{AlN}}$/*λ*→∞, the TCF of all SAW modes approaches to the TCF value of AlN material itself (~25 ppm/°C), but the M2 mode is more stable, and the dispersion amplitude is smaller overall.

### 3.2. The Influence of the Metal Electrode Thickness

According to the SAW phase velocity characteristics, i.e., the electromechanical coupling coefficient and TCF characteristics of the sandwiched structure in Figure 7, Figure 8 and Figure 9, the M2 mode not only has high $v_{p}$, but also high $K^{2}$ and good temperature stability, which is very suitable for the application of UHF devices.

In the process of SAW propagation, the wave will be reflected when the impedance does not match. To reduce the reflection, the thickness of the metal film should be minimized. However, if the metal film is too thin, the resistance of metal electrode will increase sharply, and the phenomenon of broken finger is easy to occur in the stripping process of the metal electrode preparation. Considering a compromise, the ratio between the thickness of metal film ($h_{m}$) and wavelength (*λ*) is usually selected, $h_{m}$ = 0.025~0.15. Figure 10 shows the curve characteristic of the phase velocity and electromechanical coupling coefficient of the M2 mode of the sandwiched structure, changing with $h_{\mathrm{Al}}$*/λ* when the optimal electromechanical coupling coefficient $h_{\mathrm{Al}}$*/λ* = 0.5. It can be seen that the $v_{p}$ decreases with the increase of $h_{\mathrm{Al}}$*/λ*, indicating that the increase of film thickness will reduce the operating frequency of the device; however, from the perspective of amplitude, the effect on the $v_{p}$ is not great. While $K^{2}$ increases with the increase of $h_{\mathrm{Al}}$*/λ*, about 2 percentage points changes when $h_{\mathrm{Al}}$/*λ* is improved from 0.025 to 0.15, indicating that the metal electrode thickness has a great influence on $K^{2}$.

In order to select the best electrode thickness and achieve large $K^{2}$ and $v_{p}$ at the same time, the M2 resonance effects under different electrode thicknesses are simulated, as shown in Figure 11. When $h_{\mathrm{Al}}$*/λ* = 0.05, the wave peak is the highest, the resonance effect is the best, but the corresponding $K^{2}$ value in Figure 10 is small at this time. When $K^{2}$*/λ* = 0.075, the corresponding resonant wave peak is relatively high, and the corresponding $v_{p}$ and $K^{2}$ are relatively high, respectively, at 12,520 m/s and 5.83%.

After comprehensive analysis, it can be seen that the M2 mode of sandwich structure is helpful to obtain high-performance SAW resonator. At the same time, the SAW propagation characteristics of IDT/ALN/diamond/Si traditional IDT-free surface structure resonator are compared. It can be seen that the research in this paper has great advantages. The main reason is that the phase velocity of the SAW wave is smaller with the increase of the thickness of the piezoelectric film, and the $v_{p}$ of the higher-order mode is much higher than that of the fundamental mode. However, higher-order modes can only be excited as the thickness of the piezoelectric film increases. The IDT is sandwiched in the middle of the ALN film, and the propagation speed of the excited SAW energy is equivalent to the energy of the SAW excited by 1/2 of the thickness of the piezoelectric film of the traditional I-F structure. Therefore, under the same thickness of piezoelectric film, the structure of this paper is more conducive to the excitation of higher-order modes and has higher $v_{p}$ and operating frequency. The optimized parameters are shown in Table 4.

## 4. Conclusions

A sandwiched structure AlN/IDT/diamond/Si multilayer SAW resonator is proposed. It is proved that this structure is easier to excite high-frequency resonance than that of the traditional structure. The results show that choosing appropriate geometric parameters for the sandwiched multilayer SAW structure can obtain the working frequency of up to 6.15 GHz, with a $K^{2}$ of 5.53%, $v_{p}$ of 12,470 m/s, and TCF of −6.3 ppm/°C. It is expected to guide the design and fabrication of high performance and high frequency SAW devices.

## Figures and Tables

**Figure 1 sensors-22-06479-f001:**
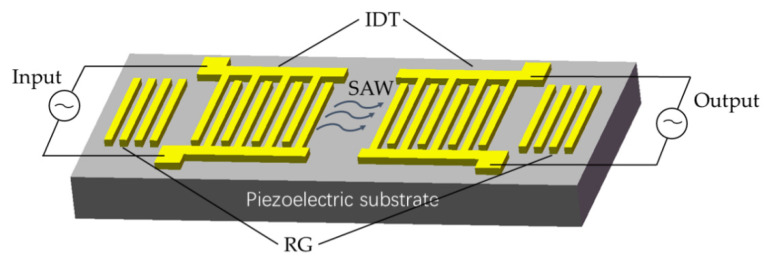
Basic structure of the SAW device [4].

**Figure 2 sensors-22-06479-f002:**
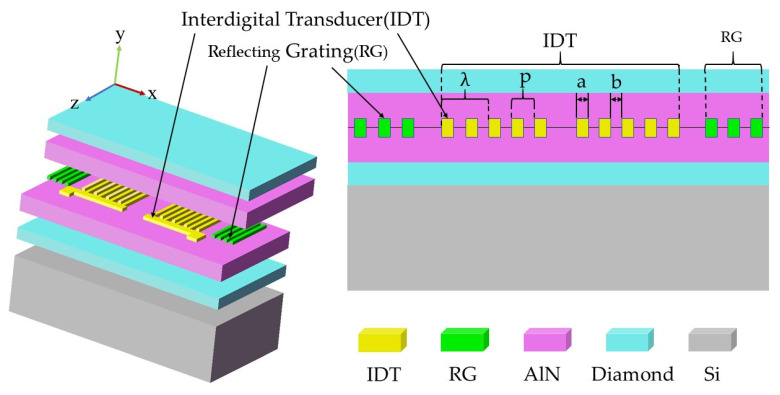
Structure diagram of sandwich SAW device.

**Figure 3 sensors-22-06479-f003:**
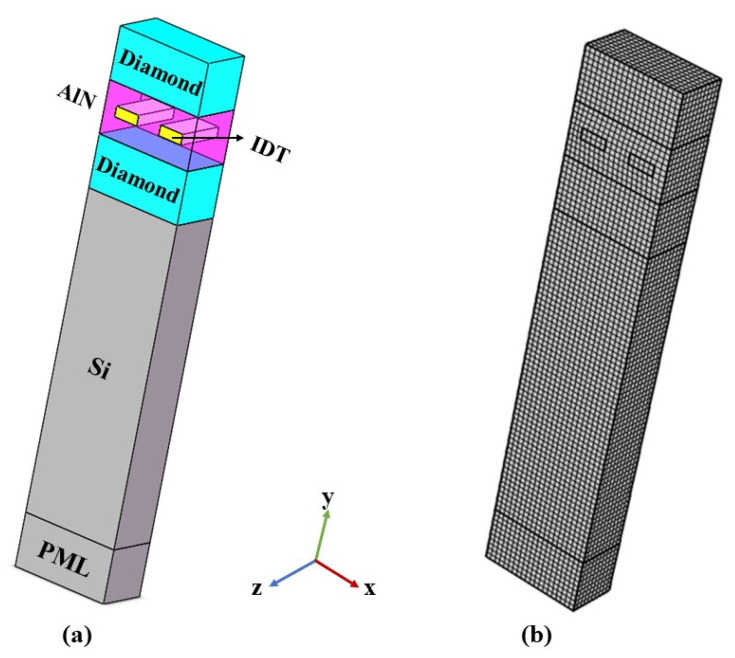
COMSOL simulation unit structure model. (**a**) Layered diagram of unit structure; (**b**) Grid division diagram.

**Figure 4 sensors-22-06479-f004:**
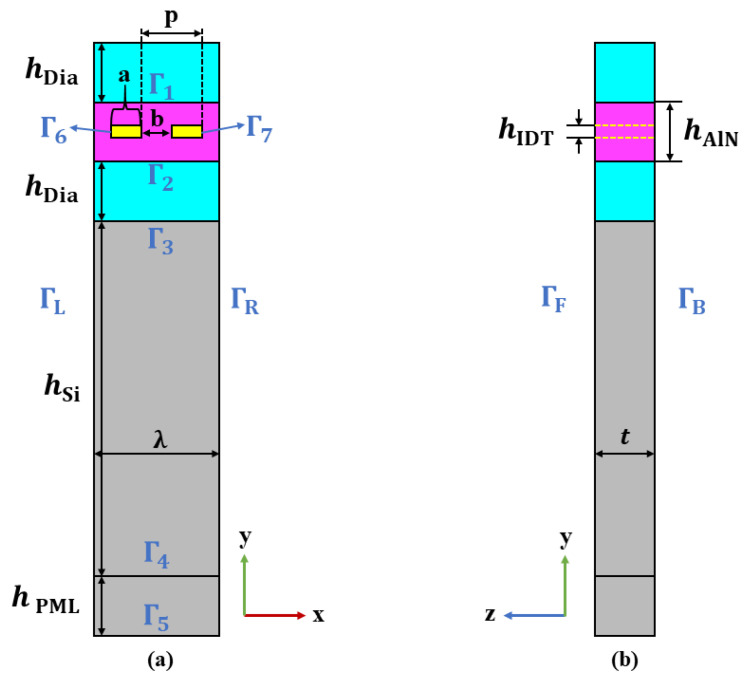
Geometric parameters and boundary conditions of element structures (**a**) planar graph of xy; (**b**) planar graph of yz.

**Figure 5 sensors-22-06479-f005:**
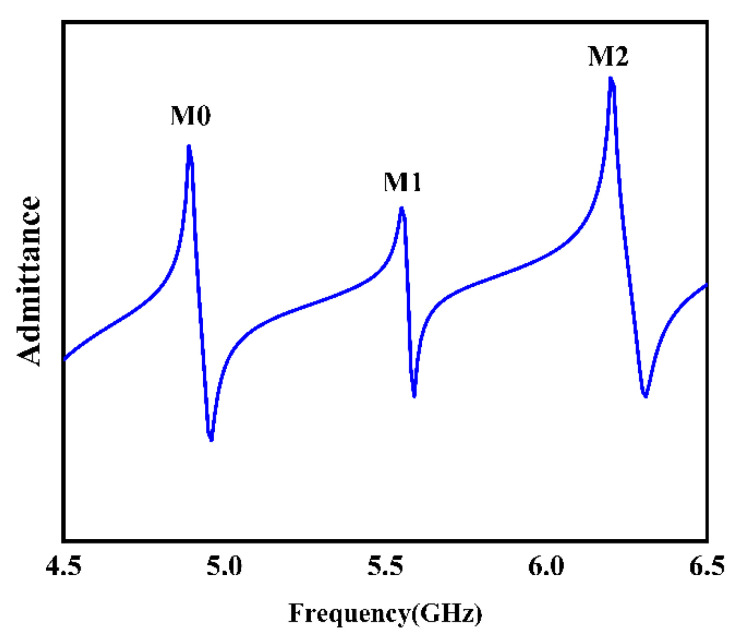
Admittance curves of sandwiched structure.

**Figure 6 sensors-22-06479-f006:**
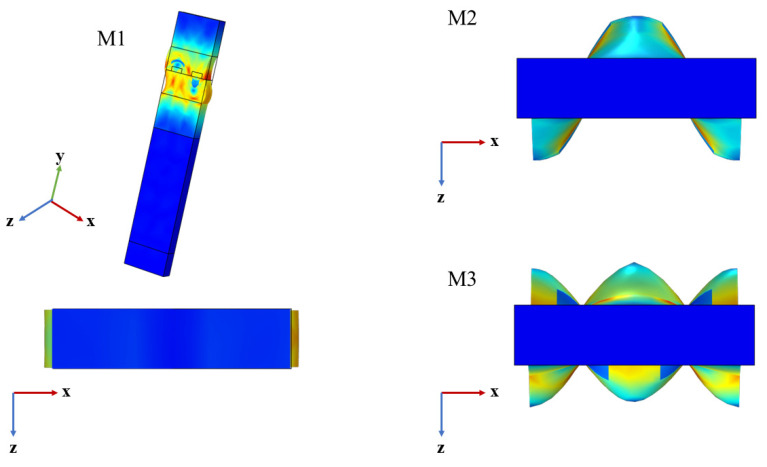
Vibration displacement diagram of three modes. (x is the SAW propagation direction, y is the vibration displacement direction of the Rayleigh wave particle, z is the vibration displacement direction of the love wave particle).

**Figure 7 sensors-22-06479-f007:**
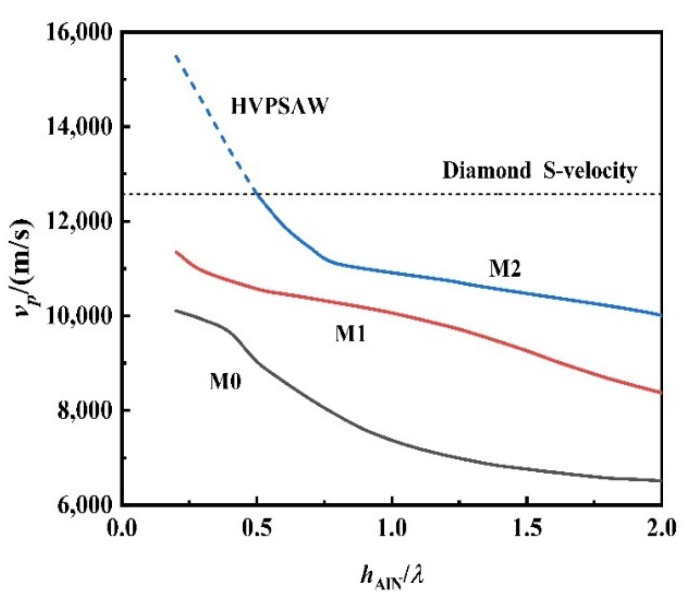
Curve characteristics of $v_{p}$ against $h_{\mathrm{AlN}}$/*λ*.

**Figure 8 sensors-22-06479-f008:**
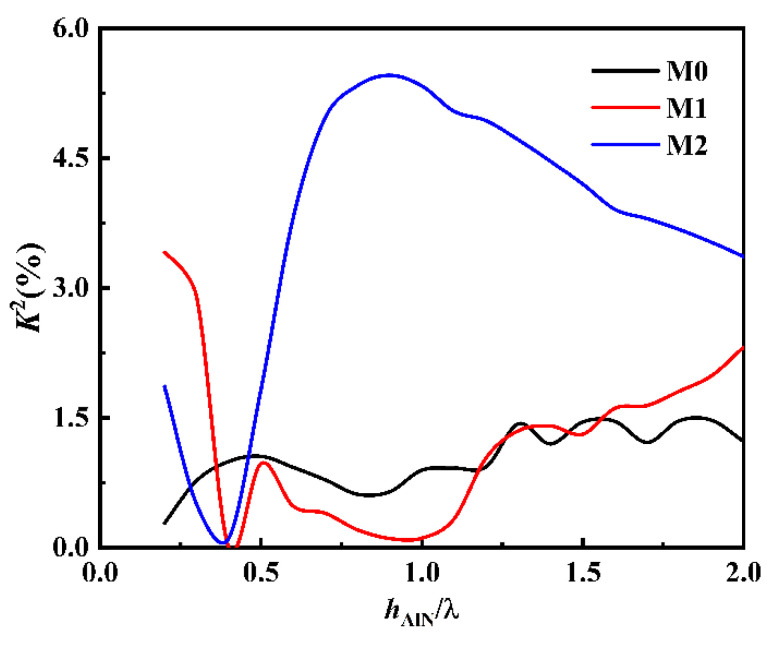
Curve characteristics of $K^{2}$ against $h_{\mathrm{AlN}}$/*λ*.

**Figure 9 sensors-22-06479-f009:**
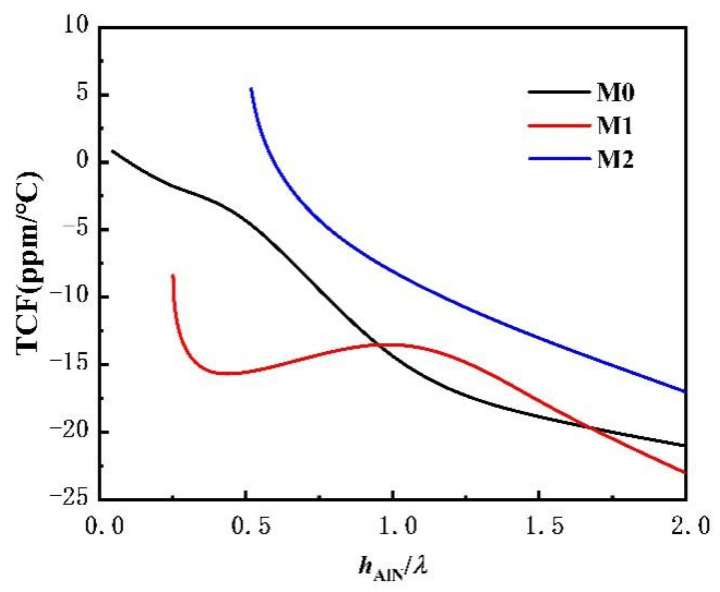
Curve characteristics of TCF against $h_{\mathrm{AlN}}$/*λ*.

**Figure 10 sensors-22-06479-f010:**
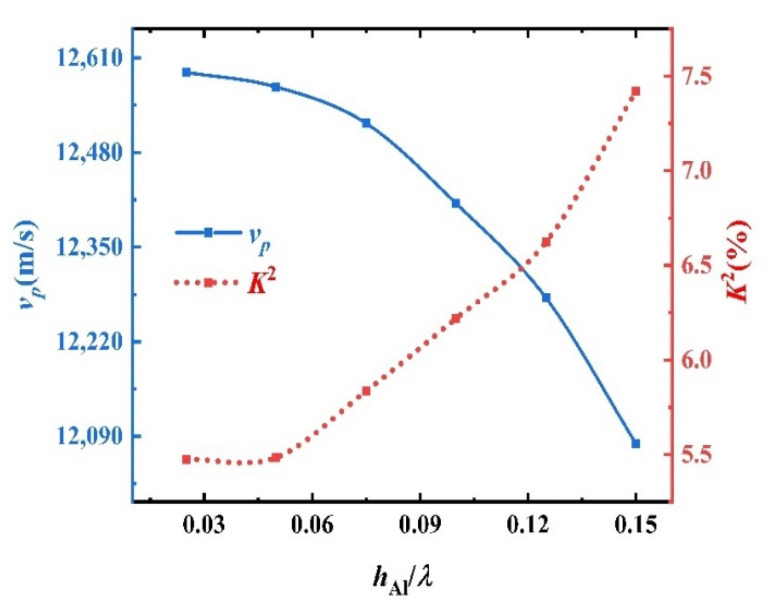
Curves of $v_{p}$ and $K^{2}$ against $h_{\mathrm{Al}}$*/λ* in M2 mode.

**Figure 11 sensors-22-06479-f011:**
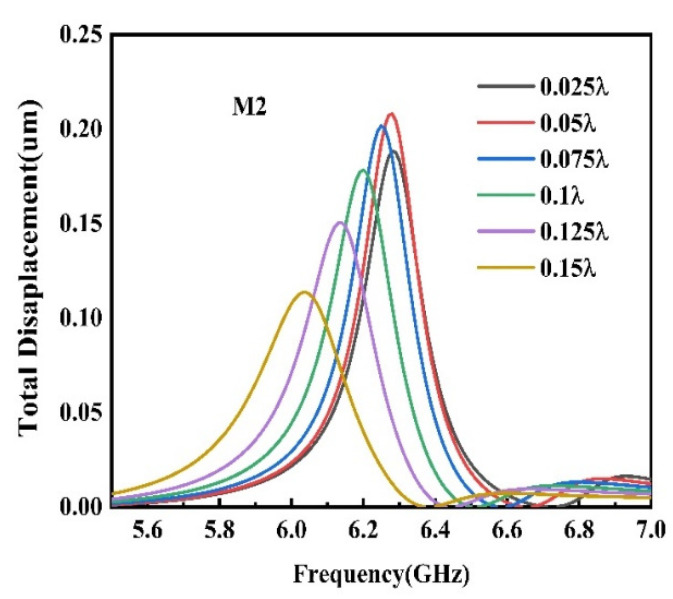
The influence of electrode thickness on resonance displacemen.

**Table 1 sensors-22-06479-t001:** Geometry characters of the FEM model.

Characters	Symbol	Values
Wavelength/μm	*λ*	2
Diamond layer thickness	$h_{\mathrm{dia}}$	0.5*λ*
Al electrode width	a	0.25*λ*
Al electrode center distance	b	0.25*λ*
AlN layer thickness	$h_{\mathrm{AlN}}$	0.5*λ*
Al layer thickness	$h_{\mathrm{Al}}$	0.1*λ*
Si layer thickness	$h_{\mathrm{Si}}$	3*λ*
PML layer thickness	$h_{\mathrm{PML}}$	0.5*λ*
Z direction thickness	t	0.5*λ*

**Table 2 sensors-22-06479-t002:** Mechanical and electrical boundary conditions.

Boundary	Mechanical Boundary Conditions	Electrical Boundary
$\Gamma_{6}$	Free boundary	1 V
$\Gamma_{7}$	Free boundary	Grounding
$\Gamma_{1},\Gamma_{2},\Gamma_{3},\Gamma_{4}$	Free boundary	Continuity
$\Gamma_{5}$	Fixed constraint	Grounding
$\Gamma_{L},\Gamma_{R},\Gamma_{F},\Gamma_{B}$	Periodic conditions	

**Table 4 sensors-22-06479-t004:** Optimized parameters of sandwiched multilayer SAW structure resonator.

Structure	Modal	$h_{\mathrm{AlN}}$	$h_{\mathrm{Al}}\;$	$f_{r}\;(\mathrm{GHz})$	$f_{\mathrm{ar}}\;(\mathrm{GHz})$	$v_{p}\;(m/s)$	$K^{2\;}(\%)$	TCF (ppm/°C)
Sandwiched structure (this work)	M2	0.9*λ*	0.075*λ*	6.15	6.32	12,470	5.53	−6.3
Traditional IDT/ALN/diamond/Si structure	M1	0.5*λ*	0.025*λ*	1.26	1.27	7620	1.55	/

## Data Availability

Not applicable.

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
