# Peer review of "High-Frequency Surface Acoustic Wave Resonator with Diamond/AlN/IDT/AlN/Diamond Multilayer Structure"

_sensors, 2022, doi:10.3390/s22176479_

Round 1

Reviewer 2 Report

The exploration of new SAW resonator with high frequency is carried out in this manuscript from view of FEM simulations. A new configuration, i.e., Diamond/AlN/IDT/AlN/Diamond with Si as the substrate is proposed, and simulated results indicate that the M2 mode in this structure has better performance. Meanwhile, the structural parameter on this mode is systematically investigated. Totally speaking, the whole paper is well organized, which focuses on the scope of Sensors. In my opinion, it can be considered to be published. Before that, some revisions must be improved:

(1) Neither theoretical analysis nor experimental measurement is included in this paper. How to verify the correctness of the numerical simulation results? The correctness of the present FEM results should be validated before the systematic numerical investigations.

(2) Which module of COMSOL Multiphysics software is used for simulation? How about the element size and mesh quantity? The necessary information should be added in this manuscript.

(3) In this paper, the thickness of PML layer is 0.5l, why? Is 0.5l enough for simulate the infinite boundary condition?

(4) The coordinate should be labeled in Fig. 3. The mode shape in Fig. 4 is shown from the three-dimensional view. However, the model in Fig. 3 is two-dimensional. They are not matched with each other.

(5) Numerical results indicate that the M2 mode can excite an operation frequency up to 6.15 GHz with electromechanical coupling coefficient of 5.53% and temperature coefficient -6.3ppm/°C. Authors should reveal the reason or underlying mechanism related to these high performance indices.

Author Response

Please see the attachment,wish you a happy life.

Reviewer 3 Report

This paper reports a diamond/AlN/IDT/AlN/Diamond based SAW resonator with high frequency. Such layered structure is one of the interesting areas of research for SAW devices, and some numerical simulation results are obtained. However, some improvements are required as suggested below:

1. The title of the article is not appropriate. The high frequency is mainly due to the high acoustic velocity material AlN and diamond rather than sandwiched IDT.

2.English writing requires extensive editing especially section 1 (introduction). The author's superscript format in the manuscript needs to be unified, such as marking or not marking the affiliation. The keywords are misspelled and too simple to highlight the focus of the paper.

3.The introduction is poorly written. The paragraph 1~2 of the introduction seems to be redundant, and it is not clear how this work is different from previously published research.

4.Again, in the introduction part, some representations are inaccurate and need to be corrected and checked to avoid repeated description and irregular writing. For example, speed of sound or sound speed or velocity, and cross finger or IDT finger etc. In addition, the abbreviations of professional vocabulary have been given in the manuscript. It is more appropriate and concise to use abbreviations habitually when describing the research content.

5. It is said that “3D modeling is selected to simplify the model”, which is quite unclear. If the model is not correct, the simulation results are not reliable. Please add the details of the 3D model and its boundary conditions.

6. Equation should be checked throughout the paper. There are many errors in Equation, for example, Equation (1) and (2): the indices of some parameters are incorrect and the subscripts have no value. The relationship between the electric field and the electric potential is not described in the formula derivation process. Furthermore, the interpretation of the φ and ψ parameters is missing from equations (3) and (4).

7. In Fig.2, the model structure should be given a specific coordinate system in order to clearly determine the direction of acoustic wave propagation.

8.In Fig.4, the vibration mode diagram cannot provide a reasonable explanation where the displacement of M1 and M2 modes along the z direction is close to zero. In addition, the clarity is not enough to distinguish the specific structure and displacement distribution.

9. In the 9th line of section 3 and the 7th line of section 3.1, the acoustic wave mode studied in the paper is described incorrectly, which is fatal. M1 is LOVE wave, Rayleigh wave, or Sezawa wave? In Table 2, the value of K2 is 1%? Please check carefully.

10.The 2nd sentence of the 2nd paragraph of section 3.1, it’s three excitation modes. And the last sentence of the paragraph “in Figure 3” is inconsistent with the content.

11. In Fig. 7, the legend format is inconsistent with other figures. And the “Figure or Fig.” need a uniform format for the manuscript.

12. What’s the Q value? The last paragraph of section 3.2 only describes that “the Q value is the largest”, which is not convincing. Please add the calculation result of the Q value and reasonable explanations

13.The metallization ratio of the electrodes is discussed in Section 3.3, but not mentioned in the last paragraph of the introduction. And it seems that the K2 and V at the metallization ratio of 0.4 are much better than 0.5~0.6 in Fig. 10. The explanation given by the author is unreasonable.

14.Some data points in Fig. 6, 7 and 10 are not accurate, and it is difficult to understand why there are obvious differences. The author needs to re-verify and carefully modify the results.

Author Response

(The authors gave the same response as above.)

Reviewer 4 Report

The article provides a theoretical analysis of a new type of SAW resonator. However, the article contains several inaccuracies and errors.

1. A typo in the title of the article (page 1, line 3) and keywords (page 1, line 16).
2. In the phrase “sub-6 and even millimeter-wave” (page 2, line 48), must be sub-6 GHz. In general, this phrase is incomprehensible: 5G Frequency Range 2 uses a frequencies from 26 to 71 GHz, 1-mm wave corresponds to 300 GHz, which is too large for SAW devices. I propose to remake or delete this phrase.
3. The phrase “For the traditional SAW device, its operation frequency is below 2.5 GHz due to the limitation of the SAW velocity of the substrate.” (page 2, line 52) is incorrect. The operating frequency of the SAW device is limited by losses in IDTs and signal attenuation during propagation of the SAW itself. This is highly dependent on the waveguide material, wave mode, temperature, and may well exceed 2.5 GHz.
4. The phrase “Choosing the right material and film thickness, and using the appropriate excitation method, can excite the layered pseudo-surface wave, pseudo-surface wave is also called Leaky SAW (LSAW)[19].” (page 2, line 56) distorts the physical essence of the LSAW phenomenon. The only condition for the appearance of a leaky surface acoustic wave in the “finite layer – semi-infinite substrate” structure is the situation when the sound velocity in the layer exceeds the sound velocity in the substrate. In this case, the acoustic wave in the layer will be continuously generate a bulk wave in the substrate at a certain angle into the depth of the substrate, which will lead to wave attenuation in the layer. This is called "radiation loss" and is not related to the viscosity of the layer and substrate.
5. The section “Principle of the sandwiched SAW resonator”, (starting page 3, line 89), contains an error. The analysis of piezoelectric devices in COMSOL does not use Maxwell's equations an all, but uses a quasi-static approximation. This is due to the fact that the velocity of an acoustic wave in a medium is approximately 5 orders of magnitude less than the velocity of an electromagnetic wave in the same medium. Equations (1) - (4) does not include the magnetic field H and magnetic induction B, but only the electric potential.
6. Figure 9 has no y-axis label, so this figure is incomprehensible.

In general, the approach used for the theoretical description of the proposed device seems to be somewhat inadequate. To describe and study the process of propagation of LSAW in a multilayer structure, one can use semi-analytical methods proposed in [A.H. Fahmy and E. L. Adler. Propagation of acoustic surface waves in multilayers: A matrix description]. This will allow to calculate the wave velocity and K2. This will require orders of magnitude less computing resources. On the other hand, using FE methods like COMSOL one can model the entire device, including emitting and receiving IDTs and RGs. This will allow calculation of the transmission coefficient S12 and reflection S11, which can be measured experimentally and verified the calculation. However, this will require a more complex model than used in the article.

In addition, when calculating at such high frequencies, it is imperative to take into account the losses in the substances used (AlN and diamond), i.e. their material constants must have an imaginary part.

Author Response

(The authors gave the same response as above.)

Round 2

Reviewer 1 Report

The authors have well revised their paper. I have no further comments.

Author Response

Dear Reviewer,

Thank you very much for taking the time to review my manuscript, which has greatly improved thanks to your previous comments, thank you for your guidance.

Kind regards,
Mr. Lei

Reviewer 3 Report

Most of the comments are responded moderately, however, there are still some minor revisions that should be addressed:

1. The keywords part, “Sand-wiched interdigital transducer FEM analysis” is too complex to highlight the focus of the paper.

2. Please check line 145~146 in page 7, the frequency corresponding to the intrinsic frequency of the open-circuit electrode is far rather than fr.

3. The equation (8) for K2 calculation is still incorrect, please check the expression of “fr-far” and reference to Ref [32].

4. In paragraph 1 of Results and discussions part, it is said M1 is the Sezawa mode in line 161, but changed to LOVE wave in line 166. Please give a reasonable explanation.

Author Response

Dear Reviewer,

Thank you for your comments, your comments are of great help to our manuscript, we have revised the article according to your comments, please see the attachment.

Kind regards,

Mr. Lei

Reviewer 4 Report

After the revision, the article has become much better and can be accepted for publication. Small final clarifications:
1. In page 5, table 2, is the row “Diamond layer thickness/μm” expressed in wavelengths or in micrometers?
2. On page 12, line 248, the abbreviation IF needs to be spelled out. Is it an intermediate frequency?

Author Response

(The authors gave the same response as above.)
